# Longitudinal examination of young married women's fertility and family planning intentions and how they relate to subsequent family planning use in Bihar and Uttar Pradesh India

Ilene S Speizer [1,2] A J Francis Zavier,[3] Lisa Calhoun,[2] Priya Nanda,[4] Niranjan Saggurti,[3] Gwyn Hainsworth[5]

¹Maternal and Child Health, University of North Carolina Gillings School of Global Public Health, Chapel Hill, North Carolina, USA
²Carolina Population Center, University of North Carolina, Chapel Hill, North Carolina, USA
³Population Council, New Delhi, India
⁴Bill and Melinda Gates Foundation India, New Delhi, India
⁵Bill & Melinda Gates Foundation, Seattle, Washington, USA

**Correspondence to**
Dr Ilene S Speizer;
Ilene_speizer@unc.edu

## ABSTRACT

**Objectives** This study examines which fertility and family planning (FP) intentions are related to subsequent FP use in a sample of young, married women in India.

**Design** We use 3-year longitudinal data from married women ages 15–19 in 2015–2016 (wave 1) who are not using contraception to examine factors associated with any use of FP in 2018–2019 (wave 2).

**Setting** Data were collected in the states of Bihar and Uttar Pradesh, India.

**Participants** A representative sample of 4893 young married women ages 15–19 was surveyed in 2015–2016 and 4000 of them were found and interviewed 3 years later. This analysis focused on the 3614 young women who were not using FP at wave 1.

**Primary outcomes** This study examines FP use at wave 2 as the main outcome variable.

**Results** Multivariate analyses demonstrated that young women who wanted to delay childbearing three or more years or who did not want any(more) children at wave 1 were more likely to use contraception at wave 2. Additionally, intention to use FP in the next 12 months at wave 1 was significantly associated with FP use at wave 2 whereas unmet need at wave 1 was not significantly related to subsequent use. A combined measure of fertility desires and intention to use FP demonstrated the importance of both measures on subsequent use. Having any children and being pregnant at wave 1 were both related to FP use at wave 2.

**Conclusions** It is important to reach young, married women prior to a first pregnancy with nuanced messages addressing their fertility and FP intentions. Programmes targeting women at antenatal and postpartum visits are important for young women to help support them to use FP to address their desires to delay or limit future childbearing for the health and well-being of themselves and their children.

## INTRODUCTION

Early and closely spaced pregnancies can have long-term negative health and well-being outcomes for young mothers and their

## STRENGTHS AND LIMITATIONS OF THIS STUDY

⇒ Longitudinal data permitted identifying meaningful measures of fertility desires and family planning use intentions for young, married women.
⇒ The results from this study can be used to better inform programme targeting of the women with the greatest contraceptive needs.
⇒ Findings demonstrated that recently married young women need information and services (at the time of marriage, prior to a birth, or at antenatal, or post-partum care) to affect their agency and intentions to use contraception in the future.
⇒ The study was limited because a key measure, intention to use family planning in the next 12 months, did not correspond to the 3-year follow-up period.
⇒ Some women may have adopted a method between the two survey waves, and this was not captured with these data; this was a limitation of this analysis.
⇒ Longitudinal examination of young married women's fertility and family planning intentions and how they relate to subsequent family planning use in Bihar and Uttar Pradesh India.

children.[1] In places where the age at marriage is low and gender norms dictate immediate childbearing post marriage, understanding factors associated with young women's (and couples') use of family planning (FP) to delay and space births is helpful for understanding how to develop programmes that lead to improved individual, maternal and child health outcomes.

At the aggregate level, unmet need for FP is a policy variable, demonstrating hypothetical gaps between fertility desires and contraceptive use. Unmet need is defined as the population of fecund, sexually active women of reproductive age who report a desire to avoid childbearing 2+ years or not have any(more) children and are not using

contraception.[2] Recent studies that examine the association between unmet need, fertility desires and subsequent fertility and FP behaviours demonstrate that there is a weak link between unmet need and outcomes.[3–5] In a study using longitudinal data from Uganda, Sarnak *et al*[3] demonstrate that among non-contracepting women in 2014, those with an unmet need are slower to adopt a contraceptive method over the 4-year follow-up period than those without an unmet need. The authors also find that intentions to use contraception in the future are significantly related to contraceptive adoption.[3] Importantly, using a combined measure of unmet need and intentions to use contraception in the future, women who intend to use, whether or not they have an unmet need, have a shorter time to adoption in the follow-up period. Further, in a study among urban women from six cities in Uttar Pradesh, India, Speizer *et al*[5] demonstrate that in a 2-year follow-up period, women who did not want any(more) children were significantly less likely to have a birth than all other women; this effect was bigger among those using contraception at baseline. However, no difference in birth experience is found between women who wanted to space and women who wanted a pregnancy soon, no matter their baseline contraceptive use. These results are consistent with an earlier study from India by Roy *et al*[6] that found that over a 6-year follow-up period, many women did not adhere to their fertility preferences nor to their FP use intentions.

Recently, colleagues have proposed the importance of considering intention to use contraception in addition to unmet need to better assess actual gaps in FP use.[7 8] Moreau *et al*[7] propose a measure called current status unmet demand that removes currently pregnant and recently postpartum women and incorporates women's intentions to use contraception in the future. In aggregate estimations of women of all ages across 46 countries, adjusted unmet need is lower (13.8%) than the standard unmet need (16.4%) and only 6.7% of women have an unmet need and intend to use and 7.2% have an unmet need but do not intend to use in the future.[7] Similarly, using data from India, Panda *et al*[8] identify those women of all ages who have an unmet need and intend to use FP in the next 12 months and demonstrate that this group is important for identifying women most in need of FP.

Prior studies focus on all women of reproductive ages (ages 15–49). Less is known about the fertility desires and FP use intentions of adolescents (ages 15–19); what is known among young women typically comes from cross-sectional studies.[9–13] Young women are of particular interest since they are at the beginning of their reproductive lives and some may want to delay a first birth or space births to pursue their aspirations (eg, stay in school, work, get to know their husbands) or for better maternal and child health outcomes. It is important to examine young women in the India context where at the time of the 2019–2021 National Family Health Survey (NFHS) 16% of women ages 20–24 in Uttar Pradesh[14] and 41% of their counterparts from Bihar[15] were married before the legal minimum age of marriage of 18; these young married women have often already begun childbearing and thus may have underserved FP needs.

This study uses longitudinal data collected among adolescents from Bihar and Uttar Pradesh, India to examine how fertility desires and FP use intentions among married adolescents evolve over the adolescent and young adult years. We also determine the utility of various measures of fertility desires, unmet need and FP intentions for supporting programmes to identify young women who have the greatest need for FP and are the most likely to adopt FP in the future.

## METHODS

### Study sample

Data for this study came from two states in northern India—Bihar and Uttar Pradesh. Both states have large populations, are among the poorest in the country and are predominately rural.[16] This study used secondary data collected as part of a study led by the Population Council in India called 'Understanding the Lives of Adolescent and Young Adults,' (UDAYA) also called the UDAYA study. In 2015–2016, wave 1 data were collected among a representative sample of unmarried girls and boys (ages 10–19) and married girls ages 15–19.[17] More details on the study design can be found elsewhere[18 19] and on the project website (www.projectudaya.in). Three years later (between 2018 and 2019), wave 2 follow-up data were collected with the sample of participants who were interviewed in wave 1.[20]

This analysis focused on the sample of women ages 15–19 who were married with Gauna performed (ie, the marriage is consummated) at wave 1 who were followed in wave 2 and had information on the fertility and FP variables of focus in this analysis. The full sample of young women surveyed in wave 1 was 4893 (see table 1). A total of 4000 of these women were reinterviewed in wave 2 (81.7%) while 893 women (18.3%) were not found at wave 2. Table 1 presents the characteristics of the full sample, the analysis sample (n=4000) and the sample lost to follow-up. The women who were not interviewed at wave 2 tended to be from Uttar Pradesh (p<0.001), have no births at wave 1 (p=0.10) and be from urban areas (p<0.001). In the analysis of which young married women were using FP at wave 2, the sample was limited to married adolescents who were non-users at wave 1 (n=3616) and had full information on the explanatory variables (n=3614).

### Outcome variable

The key outcome variable was use of any contraceptive method at wave 2. Women who at wave 2 reported using any modern method (pill, IUD, injectable, implant, condom, diaphragm, foam/jelly, female or male sterilisation, female condom, or LAM) (The list of modern methods is the same as used in large surveys such as the Demographic and Health Survey and by the FP2030

**Table 1** Descriptive information on sample of married* participants at wave 1 and the longitudinal sample for analysis

| | Full sample at wave 1* (n=4893) | Wave 2 follow-up sample (n=4000; 81.7%) | No follow-up sample (n=893; 18.3%) | P value† |
|---|---|---|---|---|
| **State (%)** | | | | |
| Uttar Pradesh | 60.4 | 58.1 | 71.0 | <0.001 |
| Bihar | 39.6 | 42.0 | 29.0 | |
| **Age in years (%)** | | | | |
| 15 | 1.9 | 1.8 | 2.5 | NS |
| 16 | 6.7 | 6.9 | 5.8 | |
| 17 | 15.6 | 15.6 | 15.6 | |
| 18 | 32.5 | 32.6 | 31.8 | |
| 19 | 43.3 | 43.1 | 44.3 | |
| **Parity (%)** | | | | |
| None | 61.0 | 59.9 | 66.0 | 0.10 |
| 1 | 31.8 | 32.3 | 29.2 | |
| 2+ | 7.2 | 7.8 | 4.8 | |
| **Education (%)** | | | | |
| None | 24.8 | 24.2 | 27.6 | NS |
| 1–4 years | 5.9 | 5.7 | 7.0 | |
| 5–7 years | 16.7 | 16.7 | 17.0 | |
| 8–9 years | 26.4 | 27.0 | 23.8 | |
| 10–11 years | 12.7 | 12.7 | 12.6 | |
| 12+ years | 13.4 | 13.7 | 12.2 | |
| **Wealth (%)** | | | | |
| Lowest | 15.2 | 14.8 | 16.9 | NS |
| Low | 20.8 | 21.3 | 18.8 | |
| Middle | 23.5 | 23.3 | 24.6 | |
| High | 23.7 | 24.0 | 22.5 | |
| Highest | 16.7 | 16.6 | 17.3 | |
| **Religion (%)** | | | | |
| Hindu | 82.3 | 83.0 | 79.0 | NS |
| Muslim | 17.5 | 16.8 | 20.7 | |
| Others | 0.2 | 0.2 | 0.3 | |
| **Caste (%)** | | | | |
| Scheduled castes/tribes | 30.0 | 29.7 | 31.1 | NS |
| Other backward castes | 57.5 | 58.1 | 54.8 | |
| General | 12.5 | 12.2 | 14.1 | |
| **Residence (%)** | | | | |
| Urban | 10.8 | 9.7 | 16.1 | <0.001 |
| Rural | 89.2 | 90.3 | 83.9 | |

All values weighted using wave 1 weights.
*Sample that was married with Gauna performed.
†F-test comparing analysis sample to those lost to follow-up.
NS, not significant.

Initiative to measure the core indicators. The only difference was that emergency contraception was not asked among the methods that women could report using in the UDAYA survey) or any traditional method (rhythm method and withdrawal) were coded 1. All others were coded zero, including those who were pregnant at wave 2 and those who reported never using a method at wave 2. Among the full sample at wave 2 (n=4000), 17% were using any method and 10% were using a modern method. The analyses presented here focused on any method use; however, models were also run with the outcome of modern method use and the results were similar.

**Key independent variables**
To understand fertility desires and FP use intentions of young married women, we examined whether variables from wave 1 were associated with use of FP at wave 2, among non-users at wave 1. First, we examined wave 1 desire for a future pregnancy and the timing of that desire (see table 2 for these variables). This was categorised as: wants a pregnancy before 2 years (soon), wants in 2 years, wants in 3 years, wants in 4+ years, undecided and wants no more children. Second, we included the standard measure of unmet need (no vs yes), calculated based on questions about fertility desires, FP use and intentionality of a recent/last pregnancy as traditionally defined.[2] We also included two questions on intention to use FP— one about intention to use in the next 12 months, and the other about intention to ever use in the future; both of these were coded as no, yes or don't know. For the intention to use FP questions, women who had ever used a method and were current non-users were not asked this question (n=75); for this analysis, these women were coded as intends to use based on their prior experience. In models that dropped the women with missing information, the results were like those presented. In this Indian context where the norm is to have 2–3 children and then get sterilised, it was not surprising that most young women intended to ever use in the future (81% at wave 1); therefore, for this analysis, we focused on intention to use in the next 12 months, even though the follow-up period is 3 years.

In response to recommendations by Moreau et al[7] and Panda et al,[8] we created measures that jointly captured fertility desires and intentions to use contraception in the future. First, we created a combined measure based on fertility desires and intentions to use FP in the future with six distinct categories (see table 3). Second, we created a combined measure based on unmet need and intention to use FP in the future; this variable had four categories that can be seen in table 3. All women who reported that they did not know if they intended to use in the next 12 months were grouped with 'no intention to use'.

We also descriptively examined if women experienced a birth between wave 1 and wave 2 (or were currently pregnant at wave 2) and the intentionality of the recent birth or current pregnancy as reported at wave 2. This information was useful for better understanding the utility of assessing fertility desires among young, married women in Bihar and Uttar Pradesh and the fluidity of these desires.

**Table 2** Fertility and FP variables at wave 1 and wave 2 of follow-up sample of women ages 15–19 and married at wave 1

| | Wave 1 % | Wave 2 % |
|---|---|---|
| Desire for a pregnancy | | |
| Wants within 2 years | 31.3 | 20.8 |
| Wants in 2 years | 18.4 | 12.5 |
| Wants in 3 years | 15.0 | 8.9 |
| Wants in 4+ years | 12.8 | 8.0 |
| Undecided | 10.2 | 8.3 |
| Wants no more | 12.3 | 41.4 |
| Currently using any method | n=3994* | |
| No | 91.0 | 82.8 |
| Yes | 9.0 | 17.2 |
| Unmet need | | n=3965† |
| No | 51.4 | 54.3 |
| Yes | 48.6 | 45.7 |
| Currently pregnant | | |
| No | 78.8 | 83.2 |
| Yes | 21.2 | 16.8 |
| Intention to use FP ever in the future | n=3998* | |
| Yes | 81.3 | 89.8 |
| No | 14.4 | 9.1 |
| Don't know | 4.3 | 1.2 |
| Intention to use FP in next 12 months | n=3998* | |
| Yes | 48.2 | 62.2 |
| No | 46.1 | 36.9 |
| Don't know | 5.7 | 0.9 |
| Experience birth or pregnancy between waves (includes currently pregnant at wave 2) | NA | |
| No | | 13.32 |
| Yes | | 86.7 |
| Intentionality of most recent birth or pregnancy experienced | NA | n=3461 |
| Wanted then | | 65.6 |
| Wanted later | | 31.2 |
| Wanted no more | | 3.2 |

Unweighted number of observations shown was 4000 unless shown otherwise.
*Two women missing information on this measure at wave 1.
†Thirty-five women missing relevant information for wave 2 unmet need measure. Wave 1 and wave 2 values used wave 2 weights that adjusted for lost to follow-up.
FP, family planning; NA, not available.

**Table 3** Fertility and family planning (FP) variables of non-users with full information at wave 1 and the correlation with wave 2 FP use

| Wave 1 characteristic | Wave 1 non-users % | Percent using FP at wave 2 | |
|---|---|---|---|
| Desire for a pregnancy | | | |
| Wants within 2 years | 33.0 | 7.9 | P<0.001 |
| Wants in 2 years | 18.6 | 12.5 | |
| Wants in 3 years | 14.3 | 16.5 | |
| Wants in 4+ years | 11.8 | 16.6 | |
| Undecided | 11.0 | 13.9 | |
| Wants no more | 11.4 | 26.2 | |
| Unmet need | | | |
| No | 46.6 | 11.6 | P=0.006 |
| Yes | 53.4 | 15.6 | |
| Intention to use FP in next 12 months | | | |
| Yes | 43.0 | 17.6 | P<0.001 |
| No | 50.8 | 9.9 | |
| Don't know | 6.3 | 18.5 | |
| Fertility desire*/FP use intention† | | | |
| Wants now/no intention | 32.2 | 9.1 | P=0.001 |
| Wants now/intention to use | 11.7 | 10.1 | |
| Wants later/no intention | 20.2 | 11.0 | |
| Wants later/intention to use | 24.5 | 18.0 | |
| Wants no more/no intention | 4.6 | 22.3 | |
| Wants no more/intention to use | 6.8 | 28.8 | |
| Unmet need/FP use intention† | | | |
| No unmet need/no intention | 30.7 | 9.8 | P=0.001 |
| No unmet need/intention to use | 15.9 | 15.3 | |
| Unmet need/no intention to use | 26.3 | 12.1 | |
| Unmet need/intention to use | 27.1 | 19.0 | |

This table focuses on the non-users of any method at wave 1 (n=3614). All percentages weighted using wave 2 weights. Significance testing determined if FP use differed by the various wave 1 fertility desire and intention to use categories.
*Wants now included those wanting in 2 years or undecided; wants later was those who want to delay 2+ years.
†No intention included those who were undecided on future use.

caste, scheduled tribe, other backward caste, general), age in years, parity (no children, one child, two or more), education level (none, primary, 8–9 years of education, 10+ years of education), religion (Hindu vs all others), wealth group (lowest, low, medium, high, highest) and place of residence (urban vs rural). The coding and distribution of these variables is shown in table 1.

## Demographic variables

Because fertility desires and contraceptive behaviours are closely tied to a woman's current pregnancy status, we adjusted for whether the woman was currently pregnant at wave 1 (no vs yes); this avoided dropping currently pregnant women from the analyses. We also adjusted for wave 1 state (Bihar vs Uttar Pradesh), caste (scheduled

## Analysis

Univariate and bivariate analyses were used to examine the associations between fertility desires and FP use intentions at wave 1 and wave 2 contraceptive use. All descriptive statistics that examined the characteristics of the wave 1 sample, including examination of loss to follow-up,

used wave 1 weights. All analyses that focused on the longitudinal sample used wave 2 weights to adjust for loss to follow-up. Multivariate logistic regression analyses were used to inform which young non-users at wave 1 were the most likely to use any FP method at wave 2, adjusting for wave 1 demographic characteristics. Recognising that some of the young women were pregnant at wave 1 and did not need FP at that time; instead of dropping them as typically done, we controlled for this wave 1 pregnancy to see how this related to FP use at wave 2. All multivariate analyses were unweighted but adjusted for sample clustering.

## Patient and public involvement

Patients and/or the public were not involved in the design or conduct or reporting or dissemination plans of this research.

## RESULTS

Table 1 shows that the sample of married adolescents was predominately ages 18 and 19 at Wave 1 (mean age is 18.0 years); this is consistent with the legal age of marriage for women in India. By wave 2, the cohort was ages 17–23 with a mean age of 20.8 years. More than half of the sample had no children and one-third had one child; the remaining women (about 8%) had two or more children by the time of the wave 1 survey. About a quarter of the sample had no education while about half had 8+ years of education. The majority of the sample was Hindu, and most participants lived in rural areas. Finally, more than half of the sample was other backward castes and a quarter were scheduled castes/tribes.

Table 2 presents a comparison of the fertility and FP variables for the longitudinal sample; all results used the wave 2 weights. At wave 1, one-third of women wanted a child soon (within 2 years) and another third wanted a child in 2 or 3 years. Nearly 10% of women were undecided about future childbearing. Thirteen per cent of women reported that they wanted to wait 4+ years before having a child (or another child) and 12% said they do not want (any)more children. By wave 2, a greater percentage of women wanted no more children as many had given birth between waves (87% experienced a pregnancy or birth). Only 9% of this young sample was using any method of contraception at wave 1 and 48.6% had an unmet need for FP. At wave 1, one-fifth of the sample was currently pregnant; these women may have a future need for FP, following the delivery of their baby.

Also included in table 2 is the intention to use FP (ever or in the next 12 months). In this India context where sterilisation is common, it is not surprising that more than four-fifths intended to use FP ever in the future. Yet only 48% of these young women intended to use FP in the next 12 months.

Table 2 includes the same measures at wave 2 to examine aggregate changes in the 3-year follow-up period. As mentioned above, 87% of women had a birth between waves or were currently pregnant at the time of the wave 2 survey. Two-thirds of the births/pregnancies were wanted then and one third were wanted 'later,' that is the woman wanted to wait 2+ years before the birth. This high birth experience between waves affected the other fertility and FP measures assessed at wave 2. Notably, while contraceptive use increased between waves (9%–17%), it did not increase as much as the desire to avoid pregnancy (12%–41%).

Table 3 focuses on the non-users of contraception at wave 1 and presents their fertility and FP variables as well as how this corresponded to FP use at wave 2. Based on wave 1 fertility desires, those young women who wanted no more children were more likely to be using by wave 2 (27%) than all other fertility desire groups; this was followed by those who wanted to wait 3+ years before the next birth. Based on unmet need at wave 1, a significantly greater percentage of those with an unmet need at wave 1 were using at wave 2 (16%) compared with those without an unmet need at wave 1 (12%); however, this difference was smaller than would be expected given the importance assigned to this global indicator. As expected, a significantly greater percentage of young women who reported at wave 1 that they intended to use FP in the next 12 months were using at wave 2 compared with those who said they did not intend to use in the next 12 months; those who were unsure if they will use were also more likely to use.

Table 3 also includes a combined fertility desire and intention to use variable. This combined variable showed that both fertility desires and FP use intentions matter in the expected directions such that those who wanted to avoid or delay childbearing were more likely to be using at wave 2, especially if they intended to use in the next year. For the combined unmet need and intention to use measure, we found significant differences between the groups; however, there was less variability in use between the unmet need/intention to use groups than when fertility desires were combined with FP use intentions.

Table 4 presents the multivariable logistic regression OR and 95% CIs for the analysis of wave 2 use of any contraception among non-users at wave 1, including wave 1 fertility desires and FP intentions and adjusting for wave 1 demographics. Model 1 shows that women who at wave 1 wanted to wait 4+ years (OR 1.68; 95% CI 1.19 to 2.37) and those women who did not want any(more) children (OR 2.27; 95% CI 1.58 to 3.25) were significantly more likely to be using a method at wave 2 than women who wanted a child within 2 years. In addition, women who wanted to wait 3 years were significantly more likely to be using at wave 2. No difference was found in wave 2 use between women who were undecided about future childbearing and women who wanted a child in 2 years as compared with women who at wave 1 wanted a child soon. Also included in this model was if the woman was pregnant at wave 1; those women who were pregnant were significantly more

**Table 4** Examination of use of any contraception at wave 2 based on wave 1 characteristics and wave 1 fertility desires/FP intentions (among non-users at wave 1, n=3614)

| | Model 1 | Model 2 | Model 3 | Model 4 | Model 5 |
|---|---|---|---|---|---|
| | OR (95% CI) | OR (95% CI) | OR (95% CI) | OR (95% CI) | OR (95% CI) |
| Fertility desires | | | | | |
| Wants within 2 years (ref) | 1.0 | | | | |
| Wants in 2 years | 1.17 (0.83 to 1.66) | | | | |
| Wants in 3 years | 1.54 (1.08 to 2.20)* | | | | |
| Wants in 4+ years | 1.68 (1.19 to 2.37)** | | | | |
| Undecided | 1.33 (0.90 to 1.97) | | | | |
| Wants no more | 2.27 (1.58 to 3.25)*** | | | | |
| Intention to use FP in 12 months | | | | | |
| No (ref) | | 1.0 | | | |
| Yes | | 1.33 (1.08 to 1.64)** | | | |
| Don't know | | 1.32 (0.87 to 2.01) | | | |
| Unmet need | | | | | |
| No (ref) | | | 1.0 | | |
| Yes | | | 1.16 (0.96 to 1.40) | | |
| Fertility desire/FP use intention | | | | | |
| Wants now/no intention (ref) | | | | 1.0 | |
| Wants now/intention to use | | | | 0.98 (0.68 to 1.42) | |
| Wants later/no intention | | | | 1.08 (0.82 to 1.42) | |
| Wants later/intention to use | | | | 1.42 (1.08 to 2.64)* | |
| Wants no more/no intention | | | | 1.69 (1.08 to 2.64)* | |
| Wants no more/ intention to use | | | | 2.20 (1.55 to 3.13)*** | |
| Unmet need/FP use intention | | | | | |
| No unmet/no intention (ref) | | | | | 1.0 |
| No unmet/intention | | | | | 1.36 (1.01 to 1.83)* |
| Unmet need/no intention | | | | | 1.19 (0.92 to 1.55) |
| Unmet need/ intention | | | | | 1.43 (1.09 to 1.86)** |

Note: All models also controled for wave 1 pregnancy status, wave 1 parity, state, caste, age, education, religion, place of residence and wealth. Multivariate models were unweighted but adjusted for sample clustering.
*P<0.05, **p<0.01, ***p<0.001.
FP, family planning.

likely to use at wave 2 than those who were not pregnant. Further, women who had no children at wave 1 were significantly less likely to use at wave 2 and those with 2+ children were significantly more likely to use at wave 2 compared with women with one child. The other wave 1 demographics (age, education, caste, religion, place of residence and wealth quintile) were in the expected directions. Findings for wave 1 pregnancy, parity and the demographics are consistent across all models shown (contact first author for these results).

Model 2 presents the same model with intention to use contraception in the next 12 months as the key variable. Adjusting for demographic factors, those women who at wave 1 intended to use FP soon were significantly more

**Table 5** Pregnancy experience between waves and intentionality of pregnancy among those who experienced a pregnancy/birth at wave 2

| Wave 1: desire for a pregnancy | Intentionality of birth/pregnancy at wave 2 (%) (n=3461) | | | | |
|---|---|---|---|---|---|
| | % had birth or pregnancy between waves | Wanted then | Wanted later | Did not want | Total % (n) |
| Wants within 2 years | 84.5 | 72.9 | 26.3 | 0.9 | 100 (n=996) |
| Wants in 2 years | 92.2 | 65.3 | 32.1 | 2.7 | 100 (n=659) |
| Wants in 3 years | 88.7 | 61.2 | 37.1 | 1.7 | 100 (n=542) |
| Wants in 4+ years | 82.0 | 57.6 | 38.4 | 4.1 | 100 (n=424) |
| Undecided | 93.4 | 66.5 | 28.6 | 4.9 | 100 (n=432) |
| Wants no more | 81.0 | 60.1 | 30.0 | 9.9 | 100 (n=408) |
| | P<0.001 | P<0.001 | | | |

Uses wave 2 weights. For percent that had a birth, n=4000 women married at wave 1 that also have wave 2 data.

likely to use at wave 2 (OR 1.33; 95% CI 1.08 to 1.64) than those who did not intend to use FP in the next year.

Model 3 includes the unmet need measure at wave 1. There was no difference in use at wave 2 between those with no unmet need and those with an unmet need at wave 1.

Model 4 includes the combined fertility desire and intention to use FP variable. Those women who wanted to avoid childbearing at wave 1 were the most likely to use, no matter their wave 1 intention to use FP. In addition, those women who wanted to delay childbearing 2+ years and who reported an intention to use in the next 12 months were significantly more likely to be using at wave 2 than their counterparts who wanted a child soon and did not intend to use. No difference in use at wave 2 was found between those who wanted to delay childbearing 2+ years but did not intend to use and those who wanted a child soon and do not intend to use.

Finally, model 5 includes the combined unmet need and intention to use FP measure. Those women who had an unmet need and intended to use FP in the next 12 months had higher odds of using (OR 1.43; 95% CI 1.09 to 1.86) than those women with no unmet need and no intention to use. Further, women who at wave 1 had no unmet need and an intention to use were also more likely to use at wave 2 (OR 1.36; 95% CI 1.01 to 1.83) than those with no unmet need and no intention to use.

Finally, table 5 provides additional context on how fertility desires among the young women play out in actual behaviours. Among women who at wave 1 wanted a child in 2 years or were undecided about future childbearing, over 90% became pregnant or had a birth between waves. Among those who wanted to wait 4+ years or did not want any (more) children, a significantly lower percentage (82% and 81%, respectively) became pregnant or had a birth between waves. Notably, among those who wanted a child soon (within 2 years), only 84% became pregnant; this might be related to sub-fecundity among these women who may have been trying to get pregnant for a while. Also shown in table 5 is the intentionality of

experienced pregnancies. While the results were significantly different across groups (p<0.001), it is notable that most births/pregnancies experienced were considered wanted then (58%–73%) or wanted later (26%–38%); only a small percentage of pregnancies were considered unwanted and this was predominately among those who did not want to become pregnant at wave 1 (10%). This table indicates post hoc rationalisation of pregnancies among these young women who were early in their childbearing years.

## DISCUSSION

The descriptive findings from this study demonstrate that over the follow-up period, 87% of the sample had a birth between waves or was pregnant at wave 2; this suggests the importance of early childbearing among young, married couples from Bihar and Uttar Pradesh. Further, pregnancy/birth experience between waves was associated with wave 1 fertility desires; however, the majority of pregnancies experienced, no matter the wave 1 desires, were considered wanted (then or later). Prior studies from India have demonstrated that fertility desires of women of all ages (not just adolescents as included here) are fluid and there is post hoc rationalisation of pregnancies/births experienced.[5 6 21] With the data available, it was not possible to know if the women changed their fertility desires between waves, if their partner or someone else influenced their pregnancy experience, if they lacked access to contraception to meet their needs, if they did not have the agency to act on their fertility desires, or if there were other reasons for the inconsistent pregnancy/birth experience in the follow-up period.

From the data available, it appears that there were FP needs in this sample of young women. At wave 1, only 9% of women were using a contraceptive method; however, unmet need for FP was high (49%). Unmet need in the UDAYA sample was higher than in the 2015/2016 India NFHS where 22.5% of women ages 15–19 in union from Uttar Pradesh[22] and 29.6% of their counterparts from

Bihar[23] had an unmet need for FP. Further, while about 40% of these young women reported an intention to use contraception in the next 12 months, FP use only increased by about 8 percentage points between wave 1 and wave 2. Whether the gap between intention and actual use reflected fluid adoption and discontinuation of methods in the follow-up period, a lack of demand for methods (eg, concerns about side effects), or barriers to access to methods when women were in need, was difficult to assess with the data available. A recent paper by Senderowicz and Maloney[24] proposes a strategy to divide unmet need into two categories: supply side versus demand side unmet need; this would help to better assess barriers to use. Further, Panda et al[8] propose that incorporating intention to use FP into unmet need measures, as done here, may help with identifying those women with the most need in India.

In our multivariate analyses, unmet need at wave 1 was not related to wave 2 contraceptive use. This is not surprising given that unmet need is meant to be used at a population level and is an investigator determined and not woman-determined assessment of need.[2 24 25] That said, fertility desires matter such that those women who reported a desire to delay 3+ years and those women who wanted to avoid childbearing were more likely to use at wave 2 than those women who wanted a child soon. Intention to use FP in the next 12 months was also related to wave 2 contraceptive use. The combined fertility desire/intentions to use variable and the unmet need/intentions to use variable were both related to subsequent use. These results inform future measurement of fertility desires and FP intentions among young women. First, unmet need on its own is not a meaningful indicator; it is difficult to calculate, not meant to be used at an individual level, and not useful among young women who are at the beginning of their reproductive careers. That said, understanding young women's fertility desires with a more detailed measure (ie, more categories than just want soon, want later and does not want as typically used) was informative for understanding their 'plans' for future childbearing, even if they were not able to follow through on these fertility desires. Further, examining young women's intentions to use a method in the next 12 months was also a meaningful and simple indicator that can be used individually or in combination with fertility desires to help inform who is most likely to adopt a method in the future. The combined fertility desires and FP use intentions measure is simple to measure, often included in counselling discussions and can be useful for programme targeting.

Our analyses demonstrated that women who were currently pregnant at wave 1 and/or had already had one or two pregnancies by the time of the wave 1 survey were significantly more likely to use at wave 2 than all others. This reflects the norm in India that young, married couples should have a birth (or two) prior to using FP.[26 27] FP programmes seeking to reach young, married women in India should promote FP messages in the antenatal

and postpartum periods to ensure that there are no missed opportunities for provision of FP services.[28] While currently pregnant women do not have a 'current need,' they do have an imminent need and may be candidates for immediate postpartum contraception, including long-acting reversible contraception.

This study is not without limitations. An important limitation is that the intention to use FP in the next 12 months variable does not correspond to the 3-year follow-up period. Given the high intention to use ever in the future, the 12-month variable seemed more relevant for this analysis. Second, we were not able to examine the timing of adoption in the follow-up period given that calendar data were not available. Further, it was not possible to examine if women adopted and discontinued in the follow-up period to meet their fertility desires. Future studies that include calendar data on timing of adoption/use of FP as part of longitudinal data are important to overcome these limitations.

## CONCLUSION

Young women's fertility desires and FP use intentions can be used by FP programmes to identify those young women with the most need for FP and to develop programmes that support fertility transitions over the early reproductive years. In this study, the women who were the least likely to adopt an FP method by wave 2 reported at wave 1 that they wanted a child soon, did not have any children and/or did not intend to use in the future. Programmes targeting these young women should focus on demand side programming to help build young women's agency to act on their family, fertility and life course desires. This could happen through outreach or mass media programmes that have messages about the benefits of FP for couple relationship strengthening, school and work attainment, as well as for the health of the mother and the child. These types of demand creation programmes may address gaps in knowledge and access, increase intention to use in the future and increase the agency of women to seek a method when or if they so desire. This may require training frontline workers on having conversations about FP with young, recently married women, their male partners and/or other gatekeepers such as mothers-in-law as well as training them to provide contraceptive methods or refer those who want to use a method they do not offer. Over time, increased engagement with young women may lead to delayed first births and better spacing between young women's births and lead to better maternal and child health and well-being.

**Contributors** ISS conceptualised the study and performed data analyses; AJFZ provided analysis inputs, including coding of some key variables; AJFZ, LC, PN, NS and GW provided insights into the study design and analysis approach. All authors reviewed drafts of the paper and approved the final version of the paper. ISS is the author acting as the guarantor.

**Funding** This research uses data from the study on 'Understanding the Lives of Adolescents and Young Adults (UDAYA) in Bihar and Uttar Pradesh' which was collected by the Population Council. Data collection was funded by Bill & Melinda

Gates Foundation (OPP1111281) and the David and Lucile Packard Foundation (# 2014-40467). This work was also supported by the Bill & Melinda Gates Foundation (INV-009814). Under the grant conditions of the Foundation, a Creative Commons Attribution 4.0 Generic Licence has already been assigned to the Author Accepted Manuscript version that might arise from this submission. We also received general support from the Population Research Infrastructure Programme through an award to the Carolina Population Centre (P2C HD050924) at The University of North Carolina at Chapel Hill.

**Competing interests**  None declared.

**Patient and public involvement**  Patients and/or the public were not involved in the design, or conduct, or reporting, or dissemination plans of this research.

**Patient consent for publication**  Not applicable.

**Ethics approval**  This study involves human participants and was approved by University of North Carolina Institutional Review Board (#: 21-2643).For the primary data collection, ethics approval was received by the Institutional Review Board at the Population Council (protocol #698). All individuals provided consent (written was optional) to participate. This study uses secondary deidentified data and was deemed exempt from ethical review by the UNC IRB (study number above). Participants gave informed consent to participate in the study before taking part.

**Provenance and peer review**  Not commissioned; externally peer reviewed.

**Data availability statement**  Data are available in a public, open access repository. Data are available on request through the following links:For UDAYA wave 1 data: https://dataverse.harvard.edu/dataset.xhtml?persistentId=doi:10.7910/DVN/RRXQNT For UDAYA wave 2 data https://dataverse.harvard.edu/dataset.xhtml?persistentId=doi:10.7910/DVN/ZJPKW5.

**ORCID iD**
Ilene S Speizer http://orcid.org/0000-0001-6204-1316

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
