## [Reviewer comments · BMJ Open]

ARTICLE DETAILS

TITLE (PROVISIONAL)	Longitudinal examination of young married women's fertility and family planning intentions and how they relate to subsequent family planning use in Bihar and Uttar Pradesh India
AUTHORS	Speizer, Ilene; Zavier, A.J.; Calhoun, Lisa; Nanda, Priya; Saggurti, Niranjana; Hainsworth, Gwyn

VERSION 1 – REVIEW

REVIEWER	Pfizer, Anne Jhpiego, Maternal and Child Survival Program
REVIEW RETURNED	30-May-2022

GENERAL COMMENTS	This paper examines longitudinal data of young married adolescents in two states in India and seeks to answer questions about measures to predict use of contraception. The only flaw I could identify in the paper (and which indirectly affects the abstracts) concerns an omission in the discussion and/or conclusion about what seems to be the inutility of combining fertility desires and intention to use. The combined odds ratio arrive at essentially the same odds of use after 3 years as fertility intention measures alone. Given that practitioners and research often need to collect less data not more, it seems a recommendation would be not to bother combining those measures, but giving more weight to assessing fertility desires. On the other hand, if data sets only include unmet need and intention to use, then a combination of the category does generate higher predictability and may be of interest. Though it is not clear to me what practical uses can be drawn from this... In short, the paper is sound, well referenced and I have found no reason not to accept. The authors may disregard my point above if they wish.
--

REVIEWER	Dev, Rubee University of Alberta
REVIEW RETURNED	04-Jun-2022

GENERAL COMMENTS	This manuscript aims to examine the fertility desires, unmet need, and FP use intentions of adolescents aged 15-19 years old in Bihar and Uttar Pradesh states of India. While this is an important study that identifies young women with FP needs, there are certain areas that could be improved. Given below are some of the suggested ways in which the paper could be improved and hope that the authors find these suggestions useful. Abstract: - What do authors mean my Wave 2 use in the last sentence under the results of abstract? Maybe it's "FP use" at Wave 2?
---

	Introduction:  - Page 3, Line 18: Spell out the abbreviation i.e., FP when it's used for the first time. It's rather spelled in line 23 in its second use. - Page 5, Line 3-15: The last paragraph sounds more like strengths of this study. Consider stating it in a way that it highlights the main purpose of this study. Methods:  - Page 6, Outcome variable: Authors need to provide the appropriate citation for the definition used for modern contraceptives. - The methods section should use the past tense as it is a report of what was done during the course of the study. Results:  - Page 7, Line 9: Number (i.e., 21,739) indicated here is not clear to me. I am assuming this number should be the sum of data from 2015 and 2020 only. - Not sure why the results (and other sections too) are written in present tense for this paper. I believe, the results section of a manuscript is also largely written using the past tense. - Page 13, Line 37: a(nother) may be a typo? - Results section is too long and difficult to follow. Since the numbers are already presented in the tables, authors could highlight only the key findings in the text to make it clearer. Discussion:  - The first paragraph of the discussion section is not required and can be omitted. Consider starting discussion with the key findings followed by the justification of findings by comparing with other studies. - There are lots of reiteration of results, which is not required. Conclusion:  - Conclusion is too long. Authors could consider condensing it by omitting the results that are stated at multiple instances.
--	---

VERSION 1 – AUTHOR RESPONSE

Reviewer: 1

Ms. Anne Pfitzer, Jhpiego

Comments to the Author:

This paper examines longitudinal data of young married adolescents in two states in India and seeks to answer questions about measures to predict use of contraception.

The only flaw I could identify in the paper (and which indirectly affects the abstracts) concerns an omission in the discussion and/or conclusion about what seems to be the inutility of combining fertility desires and intention to use. The combined odds ratio arrive at essentially the same odds of use after 3 years as fertility intention measures alone. Given that practitioners and research often need to collect less data not more, it seems a recommendation would be not to bother combining those measures, but giving more weight to assessing fertility desires.

On the other hand, if data sets only include unmet need and intention to use, then a combination of the category does generate higher predictability and may be of interest. Though it is not clear to me what practical uses can be drawn from this...

In short, the paper is sound, well referenced and I have found no reason not to accept. The authors may disregard my point above if they wish.

Response: Thank you for your careful review of the recommendations of the paper. These are excellent points that make the paper more programmatically relevant which is important to the co-authors. We have highlighted the importance of both fertility desires and family planning use

intentions which are both key discussion points during a family planning counseling session. Changes have been made to both the abstract and the discussion section with this in mind.

Reviewer: 2

Dr. Rubee Dev, University of Alberta

Comments to the Author:

This manuscript aims to examine the fertility desires, unmet need, and FP use intentions of adolescents aged 15-19 years old in Bihar and Uttar Pradesh states of India. While this is an important study that identifies young women with FP needs, there are certain areas that could be improved. Given below are some of the suggested ways in which the paper could be improved and hope that the authors find these suggestions useful.

Response: Thank you for your careful review and thoughtful suggestions that we have addressed as described below.

Abstract:

- What do authors mean my Wave 2 use in the last sentence under the results of abstract? Maybe it's "FP use" at Wave 2?

Response: Good catch, this change has been made.

Introduction:

- Page 3, Line 18: Spell out the abbreviation i.e., FP when it's used for the first time. It's rather spelled in line 23 in its second use.

Response: This change has been made

- Page 5, Line 3-15: The last paragraph sounds more like strengths of this study. Consider stating it in a way that it highlights the main purpose of this study.

Response: This paragraph has been revised to be clearer that it is describing the purpose of the study.

Methods:

- Page 6, Outcome variable: Authors need to provide the appropriate citation for the definition used for modern contraceptives.

Response: Information has been added to clarify how modern methods were defined.

- The methods section should use the past tense as it is a report of what was done during the course of the study.

Response: This change has been made to the text.

Results:

- Page 7, Line 9: Number (i.e., 21,739) indicated here is not clear to me. I am assuming this number should be the sum of data from 2015 and 2020 only.

Response: This comment does not seem to relate to this study that does not include a sample size this large, nor include data from 2015 and 2020.

- Not sure why the results (and other sections too) are written in present tense for this paper. I believe, the results section of a manuscript is also largely written using the past tense.

Response: We have modified the results section and used the past tense as proposed.

- Page 13, Line 37: a(nother) may be a typo?

Response: This was not a typo, however, we have written it out more clearly in the current version.

- Results section is too long and difficult to follow. Since the numbers are already presented in the tables, authors could highlight only the key findings in the text to make it clearer.

Response: We feel that it is important for the text and tables to stand alone but in response to this comment, we have shortened and simplified the results somewhat to focus on the key findings.

Discussion:

- The first paragraph of the discussion section is not required and can be omitted. Consider starting discussion with the key findings followed by the justification of findings by comparing with other studies.

Response: We have made this proposed change and simplified the discussion.

- There are lots of reiteration of results, which is not required.

Response: We have modified the discussion to reduce the repetition.

Conclusion:

- Conclusion is too long. Authors could consider condensing it by omitting the results that are stated at multiple instances.

Response: We have modified the discussion and conclusions sections to shorten the conclusions but keep the key methodological and programmatic recommendations.

Reviewer: 1

Competing interests of Reviewer: I declare that I have no competing interests.

Reviewer: 2

Competing interests of Reviewer: I do not have any competing interests to declare.

Editor(s)' Comments to Author (if any):

-Please revise the title of your manuscript to include the research question, study design and setting. This is the preferred format of the journal.

Response: The title has been changed.

-Please ensure that your abstract is formatted according to our Instructions for Authors:
<http://bmjopen.bmj.com/pages/authors/#research>

Response: The abstract has been modified accordingly.

-Please consider removing the instances of data "not shown" in your manuscript, either by including the relevant supporting data, or by removing the claims made.

Response: These have been removed with one exception, the suggestion to contact the first author for the full model that includes the adjusted demographic effects.

VERSION 2 – REVIEW

REVIEWER	Dev, Rubee University of Alberta
REVIEW RETURNED	28-Jun-2022
GENERAL COMMENTS	I do not have any further comments.